# Regressive evolution of an effector following a host jump in the Irish potato famine pathogen lineage

Erin K. Zess[1,2], Yasin F. Dagdas[3], Esme Peers[2], Abbas Maqbool[2], Mark J. Banfield[4], Tolga O. Bozkurt[5], Sophien Kamoun[2]*

1 Department of Plant Biology, The Carnegie Institute for Science, Stanford, California, United States of America, 2 The Sainsbury Laboratory, University of East Anglia, Norwich Research Park, Norwich, United Kingdom, 3 The Gregor Mendel Institute of Molecular Plant Biology, Vienna, Austria, 4 Department of Biological Chemistry, John Innes Centre, Norwich Research Park, Norwich, United Kingdom, 5 Imperial College London, Department of Life Sciences, London, United Kingdom

* sophien.kamoun@tsl.ac.uk

**Data Availability Statement:** All relevant data are within the manuscript and its Supporting Information files.

## Abstract

In order to infect a new host species, the pathogen must evolve to enhance infection and transmission in the novel environment. Although we often think of evolution as a process of accumulation, it is also a process of loss. Here, we document an example of regressive evolution of an effector activity in the Irish potato famine pathogen (*Phytophthora infestans*) lineage, providing evidence that a key sequence motif in the effector PexRD54 has degenerated following a host jump. We began by looking at PexRD54 and PexRD54-like sequences from across *Phytophthora* species. We found that PexRD54 emerged in the common ancestor of *Phytophthora* clade 1b and 1c species, and further sequence analysis showed that a key functional motif, the C-terminal ATG8-interacting motif (AIM), was also acquired at this point in the lineage. A closer analysis showed that the *P. mirabilis* PexRD54 (PmPexRD54) AIM is atypical, the otherwise-conserved central residue mutated from a glutamate to a lysine. We aimed to determine whether this PmPexRD54 AIM polymorphism represented an adaptation to the *Mirabilis jalapa* host environment. We began by characterizing the *M. jalapa* ATG8 family, finding that they have a unique evolutionary history compared to previously characterized ATG8s. Then, using co-immunoprecipitation and isothermal titration calorimetry assays, we showed that both full-length PmPexRD54 and the PmPexRD54 AIM peptide bind weakly to the *M. jalapa* ATG8s. Through a combination of binding assays and structural modelling, we showed that the identity of the residue at the position of the PmPexRD54 AIM polymorphism can underpin high-affinity binding to plant ATG8s. Finally, we conclude that the functionality of the PexRD54 AIM was lost in the *P. mirabilis* lineage, perhaps owing to as-yet-unknown selection pressure on this effector in the new host environment.

**Funding:** S.K. was funded by the Gatsby Charitable Foundation, UK Research and Innovation Biotechnology and Biological Sciences Research Council (UKRI-BBSRC) and the European Research Council (ERC). M.J.B. was funded by the John Innes Foundation and UKRI-BBSRC. E.P. was funded by the Biochemical Society. The funders had no role in study design, data collection and analysis, decision to publish, or preparation of the manuscript.

**Competing interests:** The authors have declared that no competing interests exist.

## Author summary

Pathogens evolve in concert with their hosts. When a pathogen infects a new host species—an event known as a "host jump"—the pathogen must evolve to enhance infection and transmission. These evolutionary processes can involve both the gain and loss of genes, as well as dynamic changes in protein function. Here, we describe an example of a pathogen protein that lost a key functional domain following a host jump, a salient example of "regressive evolution." Specifically, we show that an effector protein from the plant pathogen *Phytopthora mirabilis*, a host-specific lineage closely related to the Irish potato famine pathogen *Phytopthora infestans*, has a new (derived) amino acid change that results in a loss of interaction with a specific host component. Thus, just like terrestrial birds that have lost the capacity to fly or cave-dwelling animals that have lost their eyesight, this effector protein has become non-functional for this particular trait.

## Introduction

What needs to happen for a pathogen to successfully infect a new host species? Whether one is considering a pathogen of animals, plants, or microbes, the process of jumping onto a novel host involves three steps: contact, infection, and transmission [1]. Overcoming the latter steps presents an acute challenge, since pathogens are finely tuned to their specific host environments [1,2]. In general, a pathogen is more likely to shift to a new host species that is closely related to the original host, as the environments—the cellular machinery and the immune defenses—tend to be more similar [1–3]. However, pathogens are sometimes able to successfully jump to novel hosts that are evolutionarily distant from their original hosts, such as the recurrent jump of the bacterial pathogen *Staphylococcus aureus* from humans to livestock species [4]. In either case, by comparing extant pathogen lineages arising from a host jump, we can better understand how pathogens evolve to enhance infection and transmission in novel host environments, *i.e.*, the process of pathogen–host specialization.

Plant–pathogen interactions provide great model systems to study host jumps [5]. Knowledge gained in these systems can contribute to the global imperative to keep crop plants healthy, inform understanding of other host–parasite interactions, and reveal the elegance of evolution in action. The molecular details of plant–pathogen interactions are well-characterized, providing a framework to study the process of specialization following a host jump in fine detail—down to the level of individual proteins, and even single amino acids. During infection, plant pathogens secrete proteins and small molecules, termed effectors, that alter host-cell structure and function to enhance infection [6]. Effectors have adapted to function inside plant cells and, as such, they rapidly evolve in response to changes in the host environment [7,8]. Effector evolution is driven by two broad pressures imposed by the plant host: effectors must maintain their ability to aid infection, as well as evade detection by the plant immune system. Pathogen effectors carry out an array of functions in plant cells, including acting as enzymes, binding host proteins, and interacting with host nucleic acids [6]. In counter-defense, plants can detect pathogen effector molecules via specialized immune receptors that directly or indirectly interact with effectors, or that can sense the way that the effectors manipulate host cell processes [9]. Following a host jump, these dual pressures influence the evolution of effector molecules, and studying orthologous effectors from closely related host-specific pathogen lineages can give us granular insight into the process of pathogen–host specialization.

There are a number of studies that have investigated the molecular evolution of plant pathogen effectors following a host jump [3]. However, very few of these studies have specifically

focused on how effectors evolve to maintain their function in the context of a new cellular environment. A recent study looked at effector–target co-evolutionary dynamics in the blast fungus *Magnaporthe oryzae* [10]. The authors identified a highly conserved effector, APikL2, that is present across all assayed host-specific pathogen lineages [10]. They found that a single naturally-occurring amino acid polymorphism in APikL2 expands the range of host targets that this effector can bind to, and conclude that the mutation is likely adaptive in the lineages where it is found [10]. Other examples of research investigating the process of host specialization at the molecular level come from the *Phytophthora* clade 1c species [11]. Although it may have evolved from a broad host-range ancestor, species in clade 1c are host-specialized and are thought to have arisen through a series of host jumps to botanically distant plant species in the Solanaceae, Caryophyllaceae, and Convolvulaceae families [12]. This clade contains the economically important plant pathogen *Phytophthora infestans*, which causes the devastating late blight disease of potatoes and can also infect other Solanum species [13,14]. Previous work has compared *P. infestans* and its clade 1c 'sister' species, *Phytophthora mirabilis*—which infects the plant *Mirabilis jalapa*, colloquially known as four o'clock flower [12]. Genomic analyses comparing *P. infestans* and *P. mirabilis* revealed signatures of selection promoting change (*i.e.*, positive selection) in a high proportion of effector genes (300 out of 796 predicted genes) [12]. For one of these effectors, the protease inhibitor EPIC1, a single polymorphism between the *P. infestans* and *P. mirabilis* effector orthologs was shown to underpin the differential activity of these effectors in their respective host environments [11].

In this paper, we aimed to understand how the *Phytophthora* effector PexRD54 evolved in the context of different host environments following the presumed clade 1c host jumps. PexRD54 has been well-characterized in *P. infestans*. PexRD54 is comprised of five tandem structural domains, termed WY-domains [15–17], that pack to form an elongated molecule [18]. During *P. infestans* infection of potato, *P. infestans* PexRD54 (PiPexRD54) is translocated inside the plant cell and binds to members of the host autophagy-related 8 (ATG8) protein family [19]. The interaction of PiPexRD54 with the potato ATG8s takes place via an interface typical of ATG8-binding proteins—PiPexRD54 has a C-terminal ATG8-interacting motif (AIM) that neatly docks into the surface of ATG8, forming a tight complex [18]. The interaction of PiPexRD54 with ATG8s has the effect of dampening the host immune response by interfering with the normal operation of the selective autophagy pathway, and is also involved in remodeling the host-pathogen interface [19,20]. Our study leverages this detailed mechanistic understanding to characterize the molecular evolution of the protein.

As part of our comparative analysis of *Phytophthora* PexRD54 effectors, we observed that the *P. mirabilis* PexRD54 (PmPexRD54) ortholog carries a polymorphism within its AIM. We hypothesized that this polymorphism would impact binding to the *M. jalapa* host ATG8s, and we tested this hypothesis using a combination of *in planta* and *in vitro* binding assays, as well as structural modelling. We found that the PmPexRD54 AIM mediates weak interactions with the *M. jalapa* host ATG8s, and conclude that the AIM sequence has degenerated in the *P. mirabilis* lineage as the result of a single amino acid polymorphism at a key position. This example of regressive evolution—where a character is lost over time—contributes to our understanding of the role that this evolutionary process plays in pathogen–host specialization.

## Results

### The *Phytophthora mirabilis* PexRD54 effector has an amino acid polymorphism at a conserved residue in its ATG8-interacting motif

To understand how the *Phytophthora* effector PexRD54 has evolved in the context of different host environments following host jumps, we examined the distribution of PexRD54 across

host-specialized species. First, we collected PexRD54-related protein sequences from *Phytophthora* strains belonging to ten phylogenetically distant species [14]. We performed a preliminary phylogenetic analysis on these sequences to identify proteins closely related to *P. infestans* PexRD54 (PiPexRD54). We found that closely related proteins were only present in strains from *Phytophthora* clade 1 species (**S1 Table**), including strains of *P. infestans*, *P. ipomoeae*, *P. mirabilis*, *P. parasitica*, and *P. cactorum*, which all have different host specificities (**S1 Fig**). Of these species, *P. infestans*, *P. ipomoeae*, and *P. mirabilis* evolved from a series of recent host jumps, and infect Solanum species, morning glory, and four o'clock flower (*Mirabilis jalapa*), respectively [12] (**S1 Fig**). We constructed a phylogeny with the twenty PexRD54-related sequences from the *Phytophthora* clade 1 species strains, finding two well-supported clades, the PexRD54 clade and the PexRD54-like clade (**Fig 1A**). This phylogeny shows that PexRD54 emerged in the common ancestor of *Phytophthora* clade 1b and 1c species, likely evolving after a duplication event (**Fig 1A**).

We performed multiple sequence analyses on the PexRD54 and PexRD54-like proteins. First, we mapped the effector translocation domain—the RXLR-dEER domain [21]—based on sequence alignment and conservation, finding that this feature was present across all

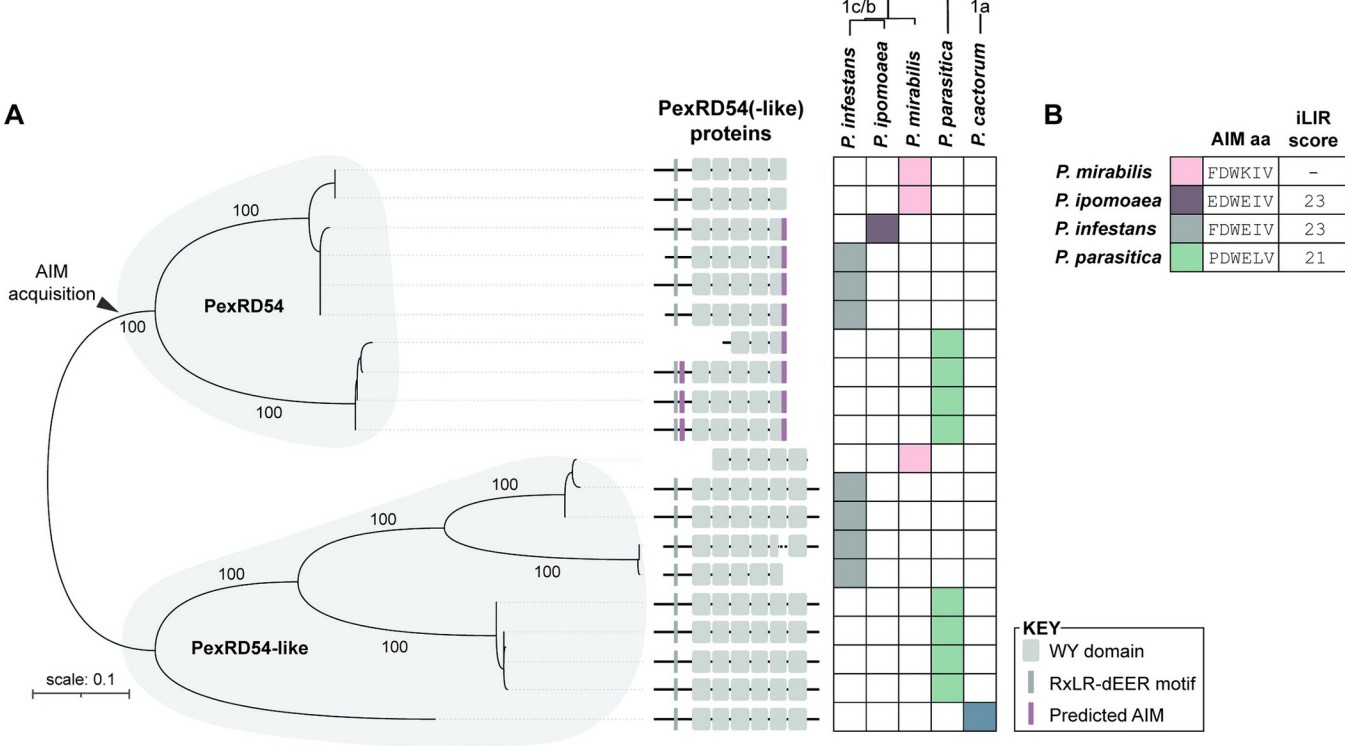

**Fig 1. The PexRD54 C-terminal AIM was acquired in the common ancestor of *Phytophthora* clade 1b and 1c species. (a)** Analysis of *Phytophthora* clade 1 PexRD54 and PexRD54-like protein sequences. Unrooted maximum-likelihood phylogenetic tree of 20 PexRD54 and PexRD54-like protein sequences (**S1 Table**) from an 285 amino acid alignment (MUSCLE [25]) spanning the PiPexRD54 first WY domain through the C-terminus, constructed using MEGA7 [26] and visualized using iTOL [27]. Protein sequences were gathered from strains of *P. mirabilis* (pink; strains P3008, P99114), *P. ipomoeae* (purple), *P. infestans* (gray; strains T30-4, KR2A1, KR2A2), *P. parasitica* (green; strains race 0, P10297, P1569, INRA-310), and *P. cactorum* (blue; strain 10300), indicated to the right. Approximate species relationships are denoted by a phylogeny adapted from Yang et al. 2017 [14] and shown in full in **S1 Fig**. The PexRD54 and PexRD54-like clades are denoted with shading, the bootstrap supports of the major nodes are indicated, and the scale bar indicates the evolutionary distance based on substitution rate. Protein representations correspond to an amino acid alignment of the full-length PexRD54 and PexRD54-like protein sequences (**S2 Fig**). Representations include predicted motifs (RxLR-dEER) and domains (WY) based on the PiPexRD54 sequence [18] and identification of key residues [16]. Predicted AIM sequences are marked in purple and were determined using the iLIR software [24]. **(b)** Table showing the PexRD54 C-terminal AIM amino acid (aa) sequences for each species and the AIM prediction score from iLIR, with more information in **S1 Table, S1** and **S2 Figs**.

full-length PexRD54 and PexRD54-like proteins (**S2 Fig**). Next, we estimated the WY-domain structure of these proteins by aligning their sequences to that of PiPexRD54, which has experimentally-validated domain definitions [18], and looked for conservation of key "WY" amino acids [16] (**S2 Fig**). Overall, we found that the tandem WY-domain structure is conserved across PexRD54 and PexRD54-like proteins, and that the PexRD54-like proteins have a single-domain C-terminal extension compared to PexRD54 proteins (**Figs 1A and S2**). Two of the proteins appear to have N-terminal truncations—a *P. parasitica* PexRD54 protein, and the *P. mirabilis* PexRD54-like protein that couldn't be resolved by further genome analysis (**Figs 1A and S2**).

Lastly, we predicted whether the proteins have ATG8-interacting motifs (AIMs) [22–24]. The core AIM sequence is composed of an aromatic amino acid followed by two amino acids and then a branched-chain amino acid, [X]-[X]-[**W/F/Y**]-[X]-[X]-[**L/I/V**], that is generally surrounded by negatively charged residues [22–23]. We used the iLIR software to predict AIMs within the PexRD54 and PexRD54-like sequences, which provides a score based on how well the amino acid residues present within a six amino acid window match with experimentally-validated AIMs [24]. We determined that most of the PexRD54 proteins have at least one predicted AIM, whereas none of the PexRD54-like proteins have predicted AIMs (**Fig 1A**; **S1 Table**). All of the PexRD54 proteins, besides those from *P. mirabilis*, have a predicted C-terminal AIM, which in *P. infestans* PexRD54 has been shown to mediate ATG8 binding [19] (**Fig 1A**; **S1 Table**). This finding indicates that the PexRD54 C-terminal AIM sequence was likely acquired in the common ancestor of *Phytophthora* clade 1b and 1c species, although it does not allow us to conclude anything about the functionality of this ancestral AIM (**Fig 1A**).

One striking observation is that the *P. mirabilis* PexRD54 C-termini does not have a predicted AIM despite its phylogenetic relatedness to AIM containing PexRD54 proteins (**Fig 1**). We decided to explore how this could reflect evolutionary pressures imposed by the *Mirabilis jalapa* host. The sequence of the *P. mirabilis* PexRD54 (PmPexRD54) AIM region (FDW<u>K</u>IV) differs from the *P. infestans* AIM (FDW<u>E</u>IV) by only one amino acid residue, the result of a single nucleotide polymorphism (**S1 Table**). In general, the PexRD54 C-terminal AIM sequences are diverse at both the nucleotide and amino acid level (**Fig 1B**; **S1 Table**). However, the central glutamate (E) residue is otherwise conserved in the PexRD54 C-terminal AIMs, and thus the lysine (K) at this position in the PmPexRD54 sequence appears to be a lineage-specific amino acid polymorphism (**Fig 1**). For AIMs, there is not a perfect relationship between what is predicted to be functional, and what is experimentally proven to be functional—in particular, some sequences that aren't predicted AIMs are shown to interact with ATG8s via the same protein-protein interaction interface as canonical AIMs [28]. We initially hypothesized that the PmPexRD54 C-terminal AIM was non-canonical, and that the lineage-specific glutamate (E) to lysine (K) polymorphism represented an adaptation to the *M. jalapa* host environment by enhancing binding to the *M. jalapa* ATG8s.

## The *P. mirabilis* PexRD54 AIM polymorphism reduces binding to *M. jalapa* host ATG8s

Following our hypothesis that the *P. mirabilis* PexRD54 (PmPexRD54) C-terminal AIM polymorphism reflects the specific selective pressures of functioning within the *M. jalapa* host environment, we explored how this residue impacts interaction with the *M. jalapa* host ATG8s (MjATG8s). First, using available transcriptomic sequence data, we confirmed that PmPexRD54 is expressed during *P. mirabilis* infection of *M. jalapa* (**S3 Fig**).

Using the same dataset, we then identified and curated MjATG8 sequences, finding six family members in total. Using phylogenetic analysis, we found that the MjATG8 isoforms

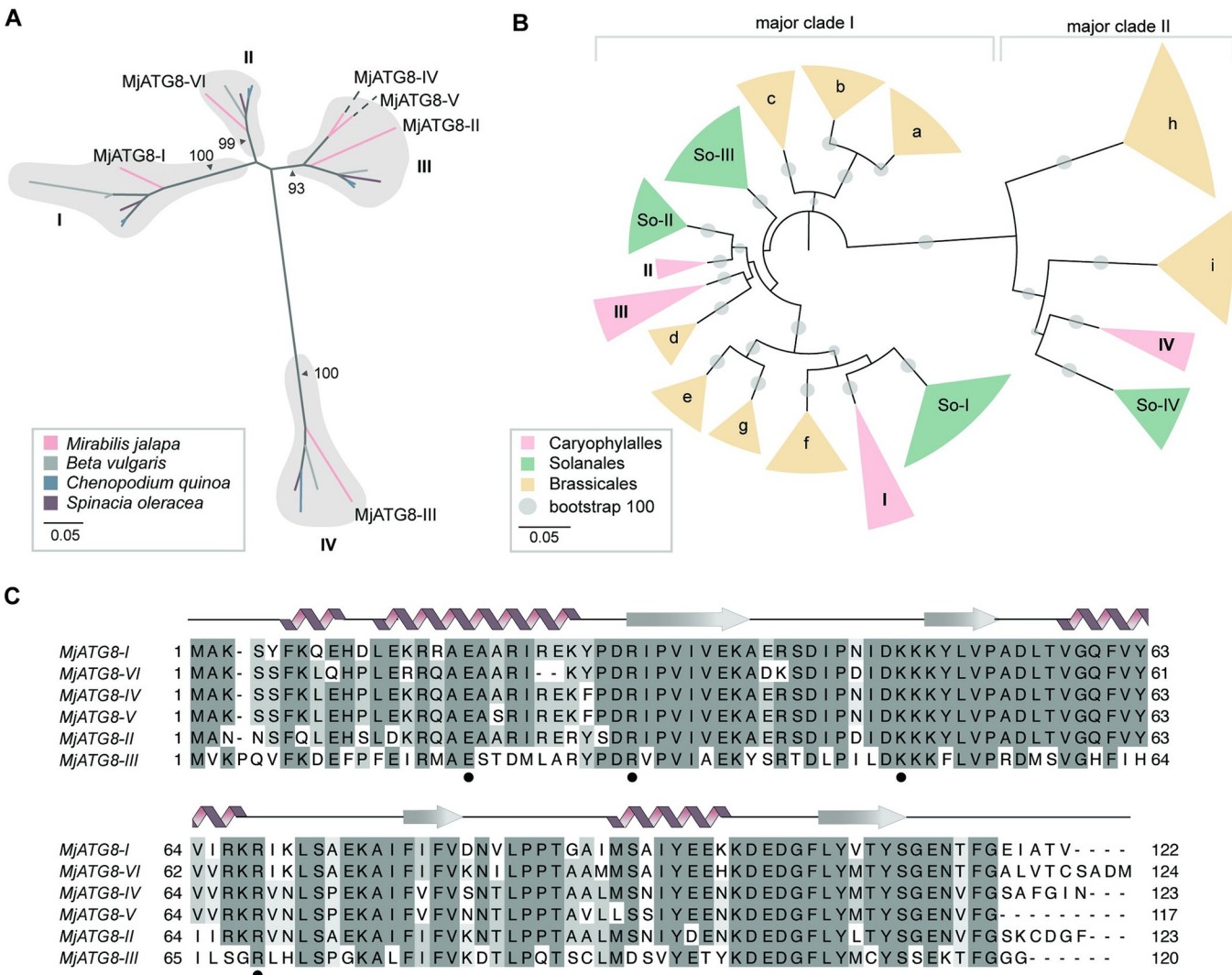

**Fig 2. *Mirabilis jalapa* ATG8s have unique evolutionary history compared to previously characterized ATG8s. (a)** Caryophyllalles ATG8 isoforms are orthologous. Unrooted maximum-likelihood phylogenetic tree of 22 ATG8 isoforms with gray shading highlighting clades, and colors indicated plant species. The tree was calculated in MEGA7 [26] from a 375 nucleotide alignment (MUSCLE [25], codon-based) and visualized using iTOL [27]. The bootstrap supports of the major nodes are indicated. The scale bar indicates the evolutionary distance based on substitution rate. **(b)** Caryophyllalles, Solanales, and Brassicales taxa have unique ATG8 subclades. Unrooted maximum-likelihood tree of 186 ATG8 isoforms, with clades collapsed based on bootstrap support and colors indicating plant order; the full tree is in the appendix, **S2 Fig**. The tree was calculated in MEGA7 [26] from a 445 nucleotide alignment (MUSCLE [25], codon-based) and visualized using iTOL [27]. The Solanales and Brassicales ATG8 clades are named following the conventions in Kellner et al. 2016 [29]. The major ATG8 clades are labelled along the top of the phylogeny. The bootstrap values of the major nodes are indicated by gray circles, with the scale as shown. The scale bar indicates the evolutionary distance based on nucleotide substitution rate. **(c)** *M. jalapa* ATG8 isoforms are sequence-diverse. Alignment of all *M. jalapa* ATG8s (MUSCLE [25]), visualized with Jalview [31], with the protein model above corresponding to the StATG8-2.2 structure, and the residues that form electrostatic contacts with AIMs are marked below (•).

cluster in four well-supported clades among other Caryophyllalles ATG8s, the plant taxa to which *M. jalapa* belongs (**Fig 2A**). Previous studies have shown that ATG8s from different plant lineages form monophyletic clades of higher taxonomic order [29]. In line with this, our phylogenetic analysis of the Caryophyllalles, Solanales, and Brassicales ATG8s shows that the Caryophyllalles ATG8s have undergone lineage-specific expansions (**Figs 2B and S2**). It has also been well-documented that plant ATG8 isoforms fall into two major clades [29], and *M. jalapa* has both of these types of ATG8s (**Fig 2B**). We also found that the MjATG8s exhibit marked sequence diversity at their N-terminus and also feature variation in regions known to

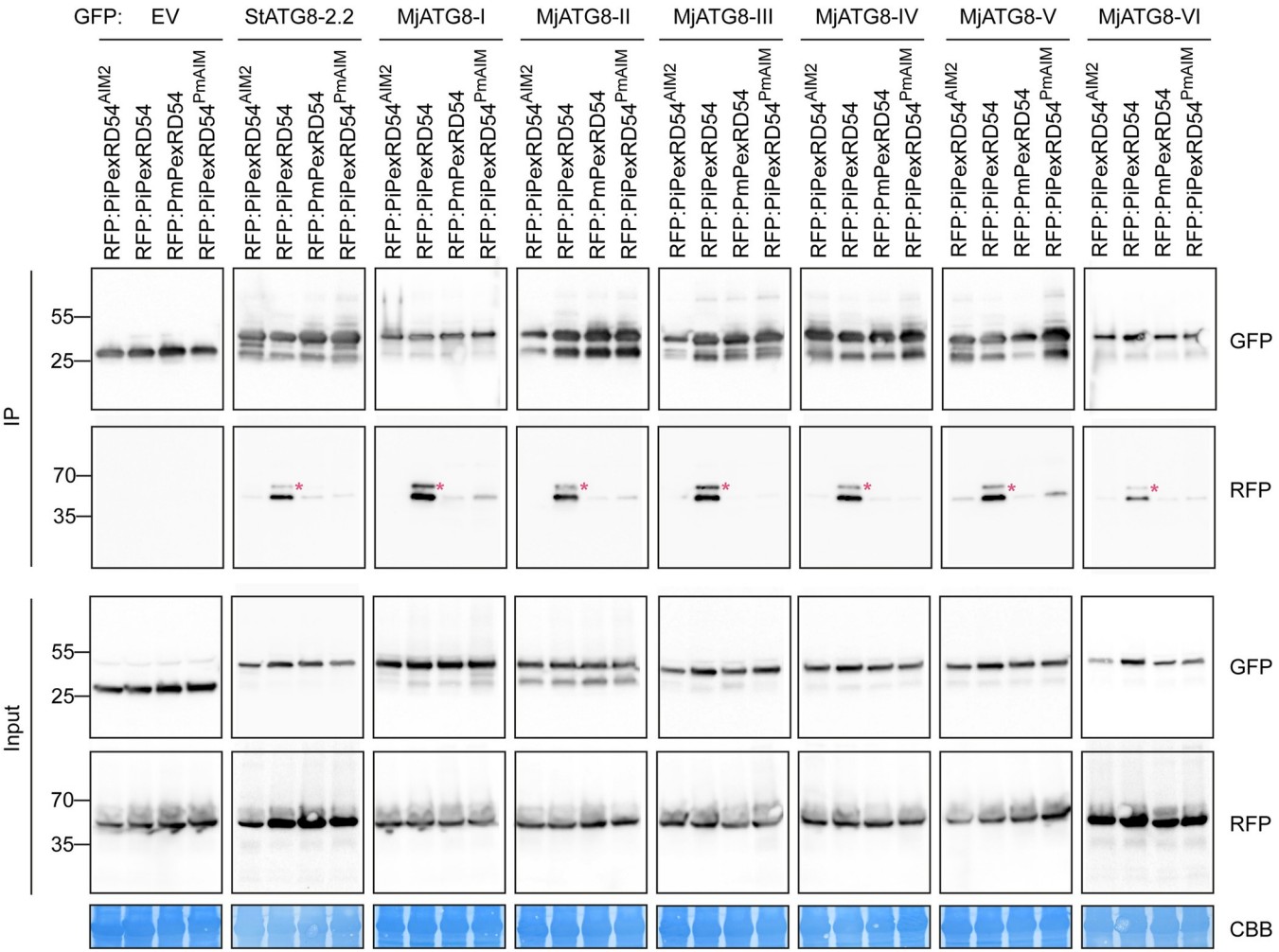

**Fig 3. The *P. mirabilis* PexRD54 AIM polymorphism reduces binding to *M. jalapa* ATG8s.** Co-immunoprecipitation experiment between PexRD54 variants (PiPexRD54^AIM2, PiPexRD54, *Pm*PexRD54, PiPexRD54^PmAIM) and *M. jalapa* ATG8s (MjATG8s). RFP:PexRD54 variants were transiently co-expressed with GFP:EV, GFP:*St*ATG8-2.2, and all GFP:MjATG8s. Immunoprecipitates (IPs) were obtained with anti-GFP antiserum and total protein extracts were immunoblotted with appropriate antisera (listed on the right). Stars indicate expected band sizes.

mediate interaction with AIM-containing proteins, such as the second β-strand [30] (**Fig 2C**). These analyses did not challenge our hypothesis that the non-canonical PmPexRD54 C-terminal AIM may enhance binding to the MjATG8s, since the lineage-specific evolution of these proteins, and sequence diversity in important binding regions, could allow for subtle structural differences that would be better targeted by the PmPexRD54 AIM.

To directly test our hypothesis, we first assayed the interaction between PmPexRD54 and the MjATG8s using *in planta* co-immunoprecipitation (co-IP). We could not detect any association between PmPexRD54 and the ATG8s, six MjATG8s and potato ATG8-2.2 (StATG8-2.2), although PiPexRD54 associated with all of the tested ATG8s (**Fig 3**). We mapped the reduction in ATG8 binding observed for PmPexRD54 to the glutamate to lysine polymorphism in the AIM, as introducing the same mutation in the PiPexRD54 background (PiPexRD54^PmAIM) abolished binding to all of the tested ATG8s (**Fig 3**). We conclude that rather than enhancing binding to the MjATG8s, the PmPexRD54 glutamate (E) to lysine (K) polymorphism reduces binding to the *M. jalapa* host ATG8s.

To quantify the reduction in binding resulting from the *P. mirabilis* PexRD54 lineage-specific polymorphism, we carried out isothermal titration calorimetry (ITC) experiments. We assayed the interaction strength between peptides matching the extended PiPexRD54 and PmPexRD54 AIM regions (10 amino acids long), respectively, and a subset of the ATG8s tested in the co-IP experiment: potato ATG8-2.2 (StATG8-2.2), MjATG8-I, and MjATG8-III. The interaction between the PiPexRD54 AIM peptide and potato ATG8-2.2 was included as a control, as this interaction has been studied extensively *in vitro* [18,19]. MjATG8-I and MjATG8-III were selected to represent the *M. jalapa* ATG8s because they are phylogenetically distant and belong to the major ATG8 clades, I and II, respectively (**Fig 2B**). We found that the PmPexRD54 AIM peptide bound weakly to all of the tested ATG8s, in each case exhibiting an affinity measurement an order of magnitude weaker than that observed for the PiPexRD54 AIM peptide (**Fig 4**; **S2 Table**).

We used two different methods to derive the thermodynamic information of these interactions. First, we individually fit the isotherm data for each technical replicate for each interaction to a single-site binding model (**Figs 4B**; **S5**; **S6** and **S2 Table**). We checked the quality of this data, noting no irregularities in the heat differences upon injection or the integrated heats of injection (**S5** **and** **S6 Fig**). We observed close agreement between the integrated heats of injection and the best fit of the data (**S5** and **S6 Fig**). The experimental replicates for each interaction also had comparable equilibrium dissociation constant ($K_D$) values (**Fig 4B**; **S2 Table**), and we observed that the values obtained for the control interaction, between the PiPexRD54 peptide and potato ATG8-2.2, were in line with previous experiments [19]. For each interaction, we also used the replicate data to perform a global analysis (**Fig 4A**). In a global analysis, the isotherms for the experimental replicates are simultaneously fit to the same binding model, producing a single, robust $K_D$ estimate for each interaction [32]. In this analysis, we found that the PmPexRD54 AIM peptide bound up to an order of magnitude weaker than the PiPexRD54 AIM peptide for all of the tested ATG8s, with PmPexRD54 binding in the low millimolar range (**Fig 4A**), similar to the analysis of the individual replicates (**Fig 4B**; **S2 Table**). These differences in binding affinity can be visually appreciated by comparing the slopes of the best fit lines for the PmPexRD54 interactions versus the PiPexRD54 interactions, with a steeper slope indicating a stronger binding affinity (**Fig 4B**). These results, both from co-immunoprecipitation and ITC, show that the *P. mirabilis* lineage-specific AIM polymorphism reduces binding to the *M. jalapa* ATG8s, suggesting that any *in vivo* interaction between PmPexRD54 and the *M. jalapa* host ATG8s during infection would likely be weak.

## The PexRD54 AIM central glutamate (E) residue is important for ATG8 binding

To better understand how the PmPexRD54 glutamate (E) to lysine (K) polymorphism leads to a reduction in binding to the *M. jalapa* ATG8s, we did additional co-immunoprecipitation experiments, as well as performed structural modelling. We found that introducing the glutamate residue back into the PmPexRD54 AIM (PmPexRD54[PiAIM])—changing the motif from FDW<u>K</u>IV to FDW<u>E</u>IV—resulted in levels of binding to StATG8-2.2 and MjATG8-I similar to those of PiPexRD54 (**Fig 5A**). These results point to the importance of the residue at this position in mediating strong ATG8 binding, since a single amino acid difference can lead to a strong gain-of-binding in the context of a full-length protein. We also looked at the impact of this residue on MjATG8 binding using structural modelling (**Fig 5B**). From prior work, we knew that in the context of the PiPexRD54–StATG8-2.2 interaction that the AIM glutamate residue makes electrostatic interactions with two ATG8 residues [18]. Using homology modelling, we show that the PexRD54 AIM glutamate residue would also likely make analogous

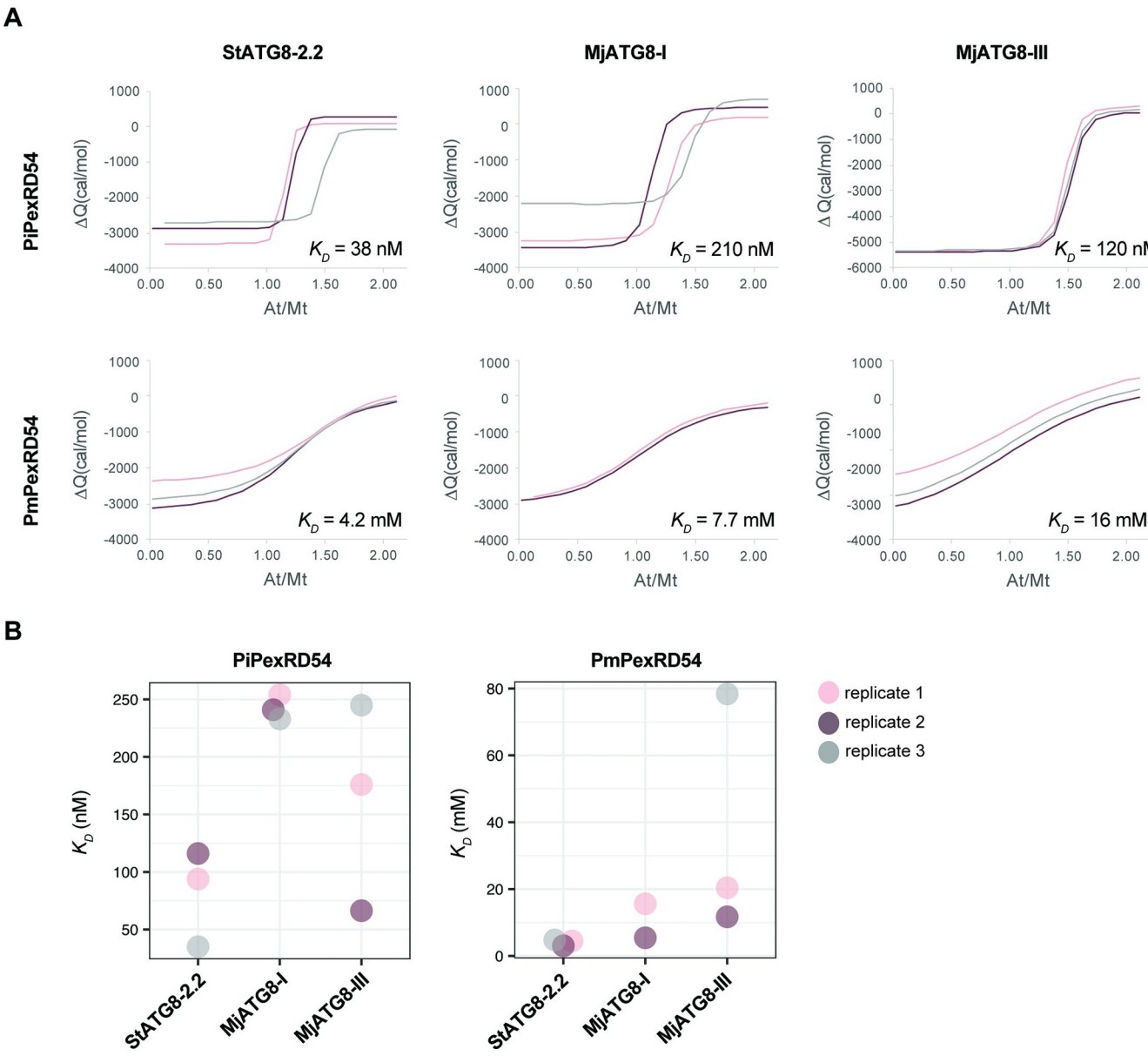

**Fig 4. *Pm*PexRD54 peptide binds weakly to ATG8s in isothermal titration calorimetry experiments.** The binding affinities between the PiPexRD54 and *Pm*PexRD54 peptides and the ATG8 isoforms ATG8-2.2, MjATG8-I, and MjATG8-III, were determined using isothermal titration calorimetry (ITC) and downstream analyses. **(a)** Global fit analysis of ITC data. The isotherms for each of the experimental replicates were simultaneously fit to the same single site binding model, producing a single robust equilibrium dissociation constant ($K_D$) estimate for each PexRD54 peptide-ATG8 interaction, using AFFINImeter analysis software [32]. $K_D$ estimates are listed in nanomolar (nM) for the PiPexRD54 interactions and millimolar (mM) for the *Pm*PexRD54 interactions. The graphs overlay the lines of best fit for the replicate isotherms (pink, grey, purple), with the integration values (ΔQ) plotted against the ratio of ligand to protein (At/Mt). **(b)** Individual fit analysis of ITC data. The isotherms for each of the experimental replicates were individually fit to a single site binding model, producing $K_D$ estimates for each PexRD54 peptide-ATG8 interaction replicate, using AFFINImeter analysis software [32]. The graphs show the $K_D$ estimates for the PiPexRD54 and *Pm*PexRD54 interaction replicates visualized using ggplot2 [33], with values in nanomolar (nM) and millimolar (mM), respectively. The graphs showing the heat differences and integrated heats of injection for each replicate are shown in **S5** and **S6 Figs**, and a table summarizing the thermodynamic information is included in **S2 Table**.

electrostatic interactions with the corresponding residues in MjATG8-I (**Fig 5B**), residues which are conserved across all *M. jalapa* ATG8s (**Fig 2C**). In contrast, because of differences in structure and charge, a lysine residue at this position in the AIM would not be able to make the same electrostatic contacts (**Fig 5C**).

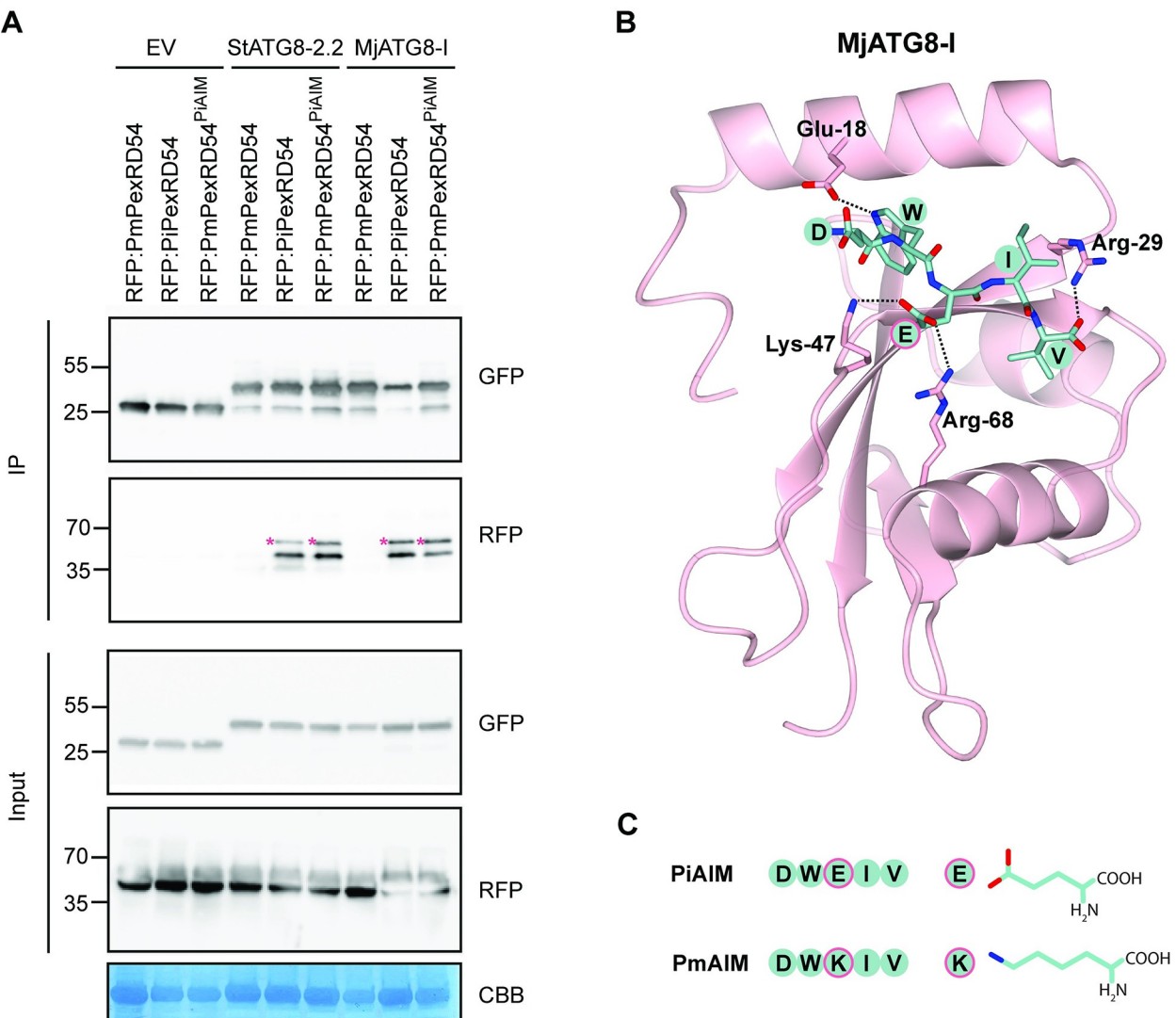

**Fig 5. The PexRD54 AIM central glutamate (E) residue is important for ATG8 binding. (a)** Co-immunoprecipitation experiment between PexRD54 variants (PmPexRD54, PiPexRD54, PmPexRD54$^{PiAIM}$) and ATG8s. RFP:PexRD54 variants were transiently co-expressed with GFP:EV, GFP:StATG8-2.2, and GFP:MjATG8-I. Immunoprecipitates (IPs) were obtained with anti-GFP antiserum and total protein extracts were immunoblotted with appropriate antisera (listed on the right). Stars indicate expected band sizes. **(b)** Homology model of MjATG8-I and PiPexRD54 AIM peptide complex viewed using CCP4 [34]. MjATG8-I and PiPexRD54 AIM are illustrated in cartoon and stick representation. Amino acids making electrostatic interactions (dashed lines) are labelled. **(c)** PiPexRD54 AIM (PiAIM) and PmPexRD54 AIM (PmAIM) amino acid sequences, including cartoon and stick representation of the differential central residue.

## Discussion

In this work, we explored how the *Phytophthora* effector PexRD54 has evolved in the context of different host environments, providing molecular insight into the process of pathogen–host specialization. Building on a detailed molecular understanding of *P. infestans* PexRD54 (PiPexRD54) function during infection of Solanum species, we investigated the evolution of characterized PexRD54 domains in orthologous effectors from closely related *Phytophthora* species that have probably arisen following host jumps [11,12,14]. We showed that PexRD54 acquired the C-terminal ATG8-interacting motif (AIM) in a common ancestor of *Phytophthora* clade 1b and 1c species, but that this motif has subsequently degenerated in the *P. mirabilis* lineage (**Fig 6**). Specifically, we found that *P. mirabilis* PexRD54 (PmPexRD54) has

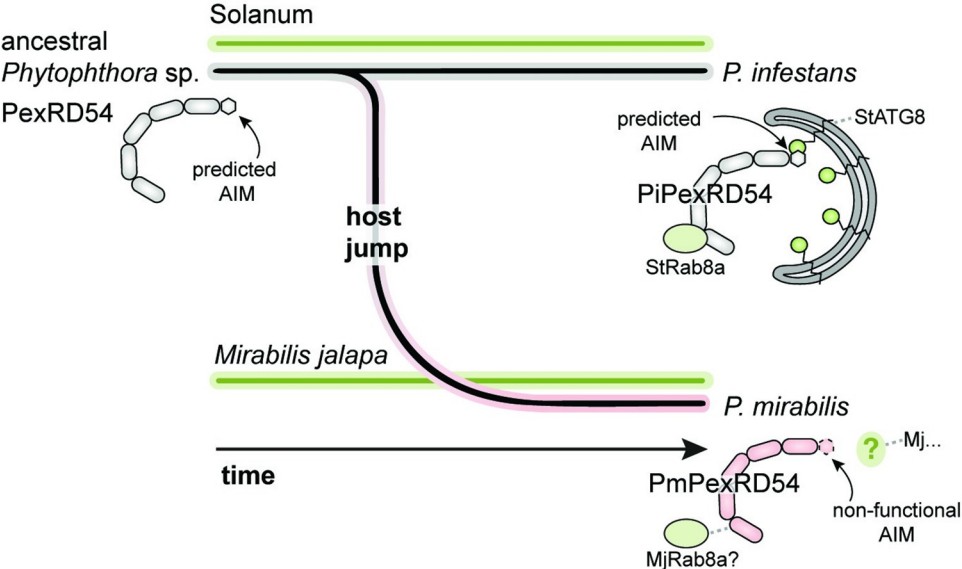

**Fig 6. Model of PexRD54 evolution following a host jump.** Schematic of the *Phytophthora* host jump from Solanum species onto *Mirabilis jalapa*, leading to the differentially specialized pathogens *P. infestans* and *P. mirabilis*. In this model, the ancestral state of the PexRD54 effector includes a predicted AIM at the c-terminus, which was maintained in the *P. infestans* lineage and lost in the *P. mirabilis* lineage. The PiPexRD54 AIM has been shown to mediate binding to the potato host ATG8s, whereas the single amino acid polymorphism in the *Pm*PexRD54 AIM precludes effector binding to the *M. jalapa* host ATG8s.

an amino acid polymorphism at a key residue within the AIM region that results in weak binding to *M. jalapa* host ATG8s (MjATG8s) (**Fig 6**). This suggests that PmPexRD54 does not directly target host ATG8s during infection, perhaps as a response to specific selection pressures imposed by the *M. jalapa* host environment.

We propose that the *P. mirabilis* PexRD54 AIM polymorphism represents an example of regressive evolution, which refers to the loss or degeneration of a trait or character [35,36]. Regressive evolution, such as loss of flight among terrestrial birds or eye loss in cave-dwelling organisms, has confounded biologists since Darwin's time [35,36]. Current debates about regressive evolution center around the role that natural selection plays in the process [35]—in short: is it involved, or not? Here, we apply this term not to a phenotypic trait or character, but to a molecular character, the PexRD54 AIM sequence. Thinking about the evolution of the *P. mirabilis* PexRD54 AIM as regressive provides a framework for understanding the mechanisms that may have led to the fixation of a polymorphism that results in the effector losing the ability to bind ATG8s. Of course, regressive evolution here refers to the ATG8 binding activity and not any other potential activities of PmPexRD54, which is presumably functioning as an virulence effector in the *M. jalapa* host. We propose several hypothetical explanations for why the PexRD54 AIM has degenerated in the *P. mirabilis* lineage which echo broader discussions about the contribution of natural selection to regressive evolutionary processes.

In line with the idea that natural selection does not drive regressive evolution, the PmPexRD54 AIM polymorphism could be the result of neutral mutation and genetic drift in a trait under relaxed selection [37]. It is possible that selective autophagy does not play the same role in plant immunity in *M. jalapa* as in Solanum species [38], and thus manipulating this pathway via direct ATG8 binding does not present an advantage, or disadvantage, to *P. mirabilis* during infection. In this case, the weak selection on the *P. mirabilis* PexRD54 AIM would allow for a polymorphism affecting functionality to become fixed by stochastic evolutionary

processes [37]. Many of the key molecular players in the selective autophagy pathway are conserved between *M. jalapa* and Solanum species, including the ATG8-interacting autophagy cargo receptor JOKA2 (also known as NBR1) [39,40]. JOKA2 has a positive role in plant immunity and, during *P. infestans* infection of potato, PexRD54 outcompetes JOKA2 for ATG8 binding to suppress the host immune response [19]. However, the selective autophagy pathway components, as well as the entire pathway, have not been characterized in *M. jalapa* or any related Caryophylalles species, and thus it is unclear whether they have the same function. Future studies investigating the role of selective autophagy in immunity in *M. jalapa*—either through reverse genetics, transgenic studies, or multiomic analyses—could help evaluate these hypotheses.

In contrast, and in line with the idea that natural selection drives regressive evolution, it is conceivable that manipulating selective autophagy via direct ATG8 binding is deleterious for *P. mirabilis* infection, and thus the PmPexRD54 AIM polymorphism represents an adaptation to the host environment. Phylogenetic analyses indicate that the *P. mirabilis* lineage probably arose from an ancestral *Phytophthora* species that infected Solanum species [11,12](**Fig 6**), and our results indicate that this ancestor likely carried a PexRD54 that could bind ATG8s with high affinity (**Fig 1**). One could speculate that, upon infecting *M. jalapa*, this ancestral PexRD54 majorly disrupted basic autophagy processes in the novel host environment, without contributing to infection, and, as a result, selection on the *P. mirabilis* lineage favored a PexRD54 allele that could not interact with ATG8s. There are numerous examples of selection operating on effector proteins, but most examples detail the evolution of effectors in response to pressure imposed by the host plant immune system [41].

In either case, one could conjecture that PmPexRD54 still retains a function in assisting *P. mirabilis* infection, despite having a non-functional AIM. A recent paper found that *P. infestans* PexRD54 has activities that are AIM-independent, including interacting with the vesicle transport regulator Rab8a [42]. The authors described that, in the context of *P. infestans* infection of Solanum species, PiPexRD54 recruits Rab8a to autophagosome biogenesis sites, thereby mimicking carbon starvation-induced autophagy [42]. These findings suggest that PmPexRD54 could retain some effector functions, such as sequestering Rab8a and targeting it to a specific cellular compartment (**Fig 6**). Moreover, we further hypothesize that PmPexRD54 is a functional effector for the simple fact that it has been maintained as an intact gene in the *P. mirabilis* genome. There are countless examples of effector loss connected to changes in host environment [43], and PmPexRD54 is even localized to a genomic compartment that has an increased incidence of gene deletions in *Phytophthora* clade 1c species [12,44]. If PmPexRD54 was no longer contributing to infection, it would seem likely that this effector would not be present or, at the very least, not expressed.

We show that a single amino acid change within the PexRD54 AIM can have a marked effect on ATG8 binding. A previous study characterized the ability of different *P. infestans* PexRD54 AIM peptide variants to bind potato ATG8-2.2 and found that the fourth position can be occupied by any amino acid, except proline, and the resulting peptide still binds in a peptide array [18]. These results contrast with our observation that the identity of the fourth PexRD54 AIM residue can underpin differential interaction with ATG8s, with the PiPexRD54 AIM (FDW<u>E</u>IV) binding very strong, and the PmPexRD54 AIM (FDW<u>K</u>IV) binding very weak. We think that the disagreement between these observations is a result of the differences in assay sensitivity, as well AIM presentation, *i.e.*, whether presented as a peptide or in the context of a full-length protein. Similarly, these same factors may influence why the PmPexRD54 peptide interacted with the tested ATG8s in isothermal titration calorimetry experiments, whereas no interaction was observed between full-length proteins in co-immunoprecipitation experiments.

In summary, we conclude that the evolution of the *P. mirabilis* PexRD54 AIM sequence is an example of regressive evolution. This is reminiscent of other plant pathogen effectors that have been lost, or have lost activity, following changes in host environments, including host jumps. These results add to a growing body of evidence that single amino acid changes can have large effects on effector functions, and that evolution is not always a process of accumulation, but sometimes of loss.

## Materials and methods

### Gene identification and cloning

The *Mirabilis jalapa* ATG8 (MjATG8) isoforms were identified using RNA sequencing datasets, amplified from cDNA using the primers listed in **S3 Table**, and cloned into the Gateway destination vector pK7WGF2. MjATG8-I and MjATG8-III were also amplified and cloned into the pOPINF vector using In-Fusion cloning [45] using the primers listed in **S3 Table**. This generated cleavable N-terminal 6xHis-tagged proteins for purification that were then transformed into the *E. coli* strain BL21 (DE3) for recombinant protein production. All constructs were verified by DNA sequencing.

PiPexRD54 (PITG_09316) and the PiPexRD54 AIM mutant (PiPexRD54$^{AIM}$) were cloned previously using Gateway cloning into the destination vectors pH7WGR2 (N-terminal RFP fusion) and PK7WGF2 (N-terminal GFP), generating the constructs RFP:PiPexRD54, RFP:PiPexRD54$^{AIM}$, GFP:PiPexRD54, and GFP:PiPexRD54$^{AIM}$ [19]. PmPexRD54 was amplified from genomic DNA of *Phytophthora mirabilis* isolate 3008 (Pm3008) using the primers listed in **S3 Table**, and cloned into the same set of Gateway destination vectors, generating RFP:PmPexRD54 and GFP:PmPexRD54. Constructs swapping the AIM sequences between PiPexRD54 and PmPexRD54—PiPexRD54$^{PmAIM}$ and *Pm*PexRD54$^{PiAIM}$—were cloned into the same Gateway destination vectors following site-directed mutagenesis, generating RFP:PiPexRD54$^{PmAIM}$, RFP:PmPexRD54$^{PiAIM}$, GFP:PiPexRD54$^{PmAIM}$, and GFP:PmPexRD54-$^{PiAIM}$. Primers in **S3 Table** were used to introduce the mutations by inverse PCR with Phusion High-Fidelity DNA Polymerase (Thermo); constructs were verified by DNA sequencing.

Gateway cloning (Invitrogen) was performed following the manufacturer's instructions. PCR-amplified sequences were cloned into the entry vector pENTR/D-TOPO and transformed into the *Escherichia coli* chemically competent cells One Shot TOP10 (Invitrogen). LR reactions were performed by mixing 0.5 μL LR Clonase II (Invitrogen), 100 ng entry clone, and 250 ng destination vector in TE buffer (pH 8.0) to a final volume of 5 μL. Reactions were incubated at room temperature for a minimum of two hours before transformation into subcloning efficiency *E. coli* DH5α chemically competent cells (Invitrogen).

In-Fusion cloning (Clontech) was performed following the manufacturer's instructions. Reactions were performed by mixing 2 μL 5x In-Fusion HD enzyme mix (Clontech), 100 ng of linearized vector, 10 ng of insert, and dH$_2$0 to a total volume of 10 μL, followed by incubation for 15 minutes at 50˚C. These reactions were transformed into subcloning efficiency DH5α chemically competent cells (Invitrogen).

### Bacterial transformation

Transformations of *E. coli* One Shot Top10 and subcloning efficiency DH5α chemically competent cells were performed according to the manufacturer's instructions (Invitrogen). Reaction products were mixed with competent cells and incubated on ice for up to 30 minutes. Cells were then heat shocked by incubation at 42˚C for 45 seconds. Immediately following, 200 μL of lysogeny broth (LB) medium was added to the cells, which were incubated at 37˚C for 45 minutes, with constant agitation. The cells were plated on LB agar plates with the

appropriate antibiotics (kanamycin or spectinomycin, 50 μg/mL) and incubated at 37˚C overnight. Transformations of *E. coli* BL21 (DE3) chemically competent cells were performed following the same protocol.

*Agrobacterium tumefaciens* strain GV3101 was used for all leaf infiltration experiments. Electroporation was performed using a cuvette with a width of 1 mm and an electroporator (Biorad) with the settings: voltage = 1.8 kV, capacitance = 25 μF, resistance = 200Ω. Immediately following electroporation, 500 μL of LB medium was added to the cells, which were then incubated at 28˚C for an hour, with constant agitation. The cells were plated on LB agar plates with the appropriate antibiotics (kanamycin 50 μg/mL and rifampicin 100 μg/mL; or spectinomycin 50 μg/mL and rifampicin 100 μg/mL) and incubated at 28˚C for approximately 48 hours.

## PCR product purification, colony PCR, and plasmid preparation

PCR products were purified using a QIAquick PCR purification kit (Qiagen). Colony PCR was performed using DreamTaq DNA polymerase according to the manufacturer's instructions (ThermoFisher Scientific). Plasmid extraction was performed using QIAprep Spin Miniprep Kit according to the manufacturer's instructions (Qiagen).

## *In planta* protein expression

Transient gene expression *in planta* was performed by delivering T-DNA constructs with *A. tumefaciens* strain GV3101 into 3–4-week old *N. benthamiana* plants as described previously [46]. *A. tumefaciens* strains carrying the plant expression constructs were diluted in agroinfiltration medium (10 mM MgCl2, 5 mM 2-[N-morpholine]-ethanesulfonic acid [MES], pH 5.6) to a final $OD_{600}$ of 0.2, unless stated otherwise. For transient co-expression assays, *A. tumefaciens* strains were mixed in a 1:1 ratio. *N. benthamiana* leaves were harvested 2–3 days after infiltration.

## Plant total protein extraction

Protein extraction was performed as described previously [46]. *N. benthamiana* leaves were ground into a fine powder in liquid nitrogen with a mortar and pestle. Ground tissue was mixed with GTEN buffer (150 mM Tris-HCl, pH 7.5; 150 mM NaCl; 10% (w/v) glycerol; 10 mM EDTA) augmented with 10mM dithiothreitol, 2% (w/v) PVPP, 1% (v/v) protease inhibitor cocktail (Sigma), and 0.2% (v/v) iGepal, at a ratio of 2x buffer volume to tissue weight. After full mixture, the samples were centrifuged at 45000 rpm at 4˚C for 30 min and the supernatants were filtered through 0.45 μM filters, resulting in the total protein extracts. For SDS-PAGE electrophoresis, total protein extracts were mixed with protein loading dye (5x final concentration: 0.2% (w/v) bromophenol blue, 200 mM Tris-HCl (pH 6.8), 2.5% (v/v) glycerol, and 4% (w/v) SDS) and incubated at 70˚C for 10 minutes before electrophoresis.

## Co-immunoprecipitation

Co-immunoprecipitation (co-IP) was carried out following the protocol described previously [46]. Immunoprecipitation was performed using affinity chromatography with GFP_Trap_A beads (Chromotek) by adding 40 μL of beads resuspended 1:1 in IP buffer (GTEN with 0.1% iGepal) to 1 mL of total protein extract, and mixing the beads and extract well by turning end-over-end for two hours at 4˚C. Samples were then centrifuged at 1000 rcf at 4˚C for 1 min; the supernatant was discarded using a needle attached to a syringe, before the beads were resuspended in 1 mL of fresh IP buffer. Samples were washed as such a total of five times before being resuspended in an equal volume of loading dye with 10 mM DTT. Elution of the proteins from the beads was performed by heating 10 minutes at 70˚C.

## SDS-PAGE electrophoresis

For western blot analysis, commercial 4–20% SDS-PAGE gels (Bio-Rad) were used for protein electrophoresis in Tris-glycine buffer (25 mM Tris, 250 mM glycine (pH 8.3), 0.1% (w/v) SDS) for approximately two hours at 120 V. For analysing *in vitro* produced proteins, commercial 16% RunBlue TEO-Tricin SDS gels (Expedeon) were used for electrophoresis in RunBlue SDS Running Buffer (Expedeon) for approximately two hours at 120 V; gels were stained with InstantBlue Protein Stain (Expedeon). For both, PageRuler Plus (Fermentas) was used as a protein size marker.

## Immunoblot analysis

Following SDS-PAGE electrophoresis, proteins were transferred onto a polyvinylidene difluoride membrane using a Trans-Blot Turbo transfer system (Bio-Rad) according to the manufacturer's instructions. The membrane was blocked with 5% milk in Tris-buffered saline and Tween 20. GFP detection was performed in a single step by a GFP (B2):sc-9996 horseradish peroxidase (HRP)-conjugated antibody (Santa Cruz Biotechnology); RFP detection was performed with a rat anti-RFP 5F8 antibody (Chromotek) and an HRP-conjugated anti-rat antibody. Pierce ECL Western Blotting Substrate (ThermoFisher Scientific) or SuperSignal West Femto Maximum Sensitivity Substrate (ThermoFisher Scientific) were used for detection. Membrane imaging was carried out with an ImageQuant LAS 4000 luminescent imager (GE Healthcare Life Sciences). SimplyBlue SafeStain (Invitrogen) staining of rubisco was used as a loading control.

## Heterologous protein production and purification

Bacteria expressing heterologous proteins were pre-cultured in 100 mL volumes of LB overnight at 37˚C with constant agitation at 180 rpm, then used to inoculate 1L volumes of autoinduction media, which were grown at 37˚C with constant agitation before being transferred to 18˚C overnight upon induction at $OD_{600}$ 0.4–0.6. Cell pellets were collected by centrifugation at 5,000 rpm for 10 minutes, before being resuspended in buffer A1 (50 mM Tris-HCl pH 8.0, 500 mM NaCl, 50 mM glycine, 5% (v/v) glycerol, 20 mM imidazole, and EDTA-free protease inhibitor). The cells were lysed by sonication and subsequently centrifuged at 18,000 rpm for 30 minutes at 4˚C to produce the clear lysate. A $Ni^{2+}$-NTA capture step produced fractions containing His-tagged protein of interest, which were concentrated as appropriate. The concentration was judged by absorbance at 280 nm, using a calculated molar extinction coefficient of each protein. For proteins with cleavable His tags (pOPINF constructs), 3c-protease was added at 10 μg/mg protein and incubated overnight at 4˚C. A final Ni2+-NTA capture step, to isolate the cleaved His tag, was followed by a final gel filtration onto a Superdex 75 26/600 gel filtration column pre-equilibrated in buffer A4 (20 mM HEPES pH 7.5, 500 mM NaCl). The fractions containing the protein of interest were pooled and concentrated as appropriate, as above. The purity of proteins was judged by running 16% SDS-PAGE gels and staining with InstantBlue (Expedeon). PexRD54 was purified as described previously [19].

## Isothermal titration colorimetry

All calorimetry experiments were recorded using a MicroCal PEAQ-ITC (Malvern, UK). To test the interaction of ATG8 proteins with PexRD54 peptides, experiments were carried out at room temperature (20˚C) in A4 buffer (20 mM HEPES pH 7.5, 500 mM NaCl). The calorimetric cell was filled with 90 μM ATG8 protein and titrated with 1 mM PexRD54 peptide. For each ITC run, a single injection of 0.5 μL of ligand was followed by 19 injections of 2 μL each.

Injections were made at 120s intervals with a stirring speed of 750 rpm. The raw titration data for the replicates of each experiment were integrated and fit to a single-site binding model using AFFINImeter software [32]. A global analysis of the interactions were performed using AFFINImeter software [32], where the isotherms for the experimental replicates were simultaneously fit to the same single-site binding model.

## Plant material

Wild-type *N. benthamiana* plants were primarily grown under glasshouse conditions, supplemented with light for a 16/8-hour light/dark cycle. For experiments testing the expression of putative ATG8-interacting proteins, *N. benthamiana* lines were grown in a controlled growth chamber with temperature 22–25˚C, humidity 45–65% and 16/8-hour light/dark cycle, due to a change a space availability.

## Phylogenetic analyses

For the PexRD54 phylogeny, protein sequences of PexRD54-related sequences were collected from *Phytophthora* strains from the species *P. infestans*, *P. parasitica*, *P. cactorum*, *P. fragariae*, *P. rubi*, *P. capsici*, *P. megakarya*, and *P. palmivora*. Using a BLAST search with relaxed parameters [47], we pulled out 62 protein sequences from the NCBI database and in-house transcriptome data. We performed a preliminary phylogenetic analysis on these sequences to identify proteins closely related to *P. infestans* PexRD54 (PiPexRD54). We constructed an unrooted maximum-likelihood phylogenetic tree of 20 PexRD54 and PexRD54-like protein sequences (**S1 Table**) from an 285 amino acid alignment (MUSCLE [25]) spanning the PiPexRD54 first WY domain through the C-terminus, constructed using MEGA7 [26], with bootstrap values based on 1000 iterations and visualized using iTOL [27]. These protein sequences were from strains of *P. mirabilis* (pink; strains P3008, P99114), *P. ipomoeae* (purple), *P. infestans* (gray; strains T30-4, KR2A1, KR2A2), *P. parasitica* (green; strains race 0, P10297, P1569, INRA-310), and *P. cactorum* (blue; strain 10300).

For the ATG8 phylogenies, nucleotide sequences of ATG8s from Solanales and Brassicales were collected from Kellner et al., 2017 [29]. The potato ATG8-2.2 sequence was used to identify the homologs from the Chenopodiaceae and Nyctaginaceae families (Order: Caryophylalles) using BLAST (NCBI) [47]. The phylogenetic tree showing all ATG8s was calculated in MEGA7 [26] from a 444-nucleotide alignment (MUSCLE [25], codon-based) with bootstrap values based on 1000 iterations and visualized using iTOL [27]. To simplify the phylogenetic tree, some branches were collapsed into clades according to the bootstrap values of the nodes; Solanales and Brassicales clades were labelled using the conventions in Kellner et al., 2017 [29]. The phylogenetic tree showing the ATG8s from Caryophylalles was calculated in MEGA7 [26] from a 372-nucleotide alignment (MUSCLE [25], codon-based) with bootstrap values based on 1000 iterations and visualized using iTOL [27].

## Homology modelling

Due to high sequence identity, ATG8-2.2 was used as a template to generate a homology model of MjATG8-1. The amino acid sequence of MjATG8-I was submitted to Protein Homology Recognition Engine V2.0 (Phyre2) for modelling [48]. The coordinates of ATG8-2.2 structure (5L83) were retrieved from the Protein Data Bank (PDB) and assigned as modelling template by using Phyre2 Expert Mode. The resulting model of MjATG8-I comprised amino acids Thr-4 to Glu-112 and was illustrated in CCP4MG software [34].

## Supporting information

**S1 Table. Characteristics of PexRD54 and PexRD54-like proteins in Fig 1.** PexRD54 and PexRD54-like proteins are listed in the same order (top-bottom) as Fig 1, with the numeric ID corresponding to S2 Fig. For each protein, the clade as determined in the Fig 1 phylogeny (PexRD54, RD54; PexRD54-like, RD54L) is listed. The *Phytophthora* species, NCBI accession, and length are also recorded for each PexRD54 and PexRD54-like protein. The number of predicted WY domains are noted, based on alignment to the PiPexRD54 sequence [18] and identification of key residues [16]. The aligned amino acid and nucleotide sequences at the PiPexRD54 AIM site are shown. The AIM prediction score from the iLIR software [25] is listed for each amino acid sequence at the PiPexRD54 AIM site, where '-' denotes no predicted AIM. (PDF)

**S2 Table. Summary of the thermodynamic and kinetic data for the isothermal titration calorimetry experiments.** Table summarizing the thermodynamic and kinetic data for the isothermal titration calorimetry experiments presented in S5 and S6 Figs. (PDF)

**S3 Table. Primers used in this study.** Table listing all primers used for cloning the constructs used in this study. Amplicon sizes marked with an asterisk (*) are dependent on the vector context. (PDF)

**S1 Fig. An overview of the phylogenetic relationships and host range of *Phytophthora* clade 1 species. (a)** Phylogeny of *Phytophthora* clade 1 species was previously reported and the tree depicted here is adapted from Yang et al. 2017 [14]. Species with available genome sequencing data are color-coded corresponding to Fig 1; species without available sequencing data are shown in grey. The *Phytophthora* subclades (1a, 1b, 1c) are noted. **(b)** Host specificity of *Phytophthora* clade 1 species. (PDF)

**S2 Fig. Full-length alignment of PexRD54 and PexRD54-like proteins from Fig 1.** A 482 amino acid alignment (MUSCLE [25]) of the full-length PexRD54 and PexRD54-like proteins from Fig 1. The proteins are listed in the same order (top-bottom) as Fig 1, with the numeric ID [1–20] corresponding to the key and to Table S1. The predicted WY domain boundaries were mapped based on the PiPexRD54 sequence (WY-1 –WY-5) [18] and identification of key residues based on the WY domain MEME (WY-6) [16]. The RxLR-dEER motif is noted, as is the location of the PexRD54 C-terminal AIM site. (PDF)

**S3 Fig. PmPexRD54 is expressed during *P. mirabilis* infection of *M. jalapa*.** Graph representing the transcript abundance for PmPexRD54 and the control elongation factor 1-alpha (EF1a) in *P. mirabilis* strain 09316 mycelia and 2–6 days post infection (dpi) of *M. jalapa*, across three technical replicates. Transcript abundance was measured by RNAseq and is reported in transcripts per million (TPM). (PDF)

**S4 Fig. Phylogenetic relationship between ATG8s from the Caryophylalles, Solanales, and Brassicales.** Unrooted maximum-likelihood tree of 186 ATG8 isoforms, with clades marked and colored as in Fig 2B. The tree was calculated in MEGA7 [26] from a 445 nucleotide alignment (MUSCLE [25], codon-based). The Solanales and Brassicales ATG8 clades are named following the conventions in Kellner et al. 2017 [29]. The bootstrap values of the major nodes

are indicated. The scale bar indicates the evolutionary distance based on nucleotide substitution rate.
(PDF)

**S5 Fig. PiPexRD54 AIM peptide interaction with StATG8-2.2, MjATG8-I and MjATG8-III in isothermal titration calorimetry.** The binding affinities between PiPexRD54 AIM peptide and StATG8-2.2, MjATG8-I, and MjATG8-III were determined using isothermal titration calorimetry (ITC). The top panels show heat differences upon injection of peptide ligands, and the lower panels show integrated heats of injection (•) and the best fit (pink line) to a single site binding model using AFFINImeter analysis software [32].
(PDF)

**S6 Fig. PmPexRD54 AIM peptide interaction with StATG8-2.2, MjATG8-I and MjATG8-III in isothermal titration calorimetry.** The binding affinities between PiPexRD54 AIM peptide and StATG8-2.2, MjATG8-I, and MjATG8-III were determined using isothermal titration calorimetry (ITC). The top panels show heat differences upon injection of peptide ligands, and the lower panels show integrated heats of injection (•) and the best fit (pink line) to a single site binding model using AFFINImeter analysis software [32].
(PDF)

## Acknowledgments

We thank Clare Stevenson, of the John Innes Centre Biophysical Analysis team, for technical assistance during isothermal titration calorimetry (ITC) experiments. We are also grateful for technical assistance provided by AFFINImeter. Lastly, we are thankful to many colleagues, especially members of the Kamoun lab, for discussions and support.

## Author Contributions

**Conceptualization:** Erin K. Zess, Yasin F. Dagdas, Tolga O. Bozkurt, Sophien Kamoun.

**Formal analysis:** Erin K. Zess.

**Funding acquisition:** Sophien Kamoun.

**Investigation:** Erin K. Zess, Yasin F. Dagdas, Esme Peers.

**Methodology:** Erin K. Zess, Esme Peers, Abbas Maqbool, Mark J. Banfield.

**Project administration:** Erin K. Zess, Sophien Kamoun.

**Resources:** Yasin F. Dagdas, Abbas Maqbool, Tolga O. Bozkurt.

**Supervision:** Sophien Kamoun.

**Visualization:** Erin K. Zess.

**Writing – original draft:** Erin K. Zess, Sophien Kamoun.

**Writing – review & editing:** Erin K. Zess, Yasin F. Dagdas, Sophien Kamoun.

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
