## [Decision Letter · Decision Letter 0]

7 Mar 2022

Dear Kamoun,

Thank you very much for submitting your manuscript "Regressive evolution of an effector following a host jump in the Irish Potato Famine Pathogen Lineage" for consideration at PLOS Pathogens. As with all papers reviewed by the journal, your manuscript was reviewed by members of the editorial board and by several independent reviewers. The reviewers appreciated the attention to an important topic. Based on the reviews, we are likely to accept this manuscript for publication, providing that you modify the manuscript according to the review recommendations.

Sincerely,

Yuanchao Wang

Associate Editor

PLOS Pathogens

Hui-Shan Guo

Section Editor

PLOS Pathogens

Kasturi Haldar

Editor-in-Chief

PLOS Pathogens

orcid.org/0000-0001-5065-158X

Michael Malim

Editor-in-Chief

PLOS Pathogens

orcid.org/0000-0002-7699-2064

Reviewer Comments (if any, and for reference):

Reviewer's Responses to Questions

**Part I - Summary**

Reviewer #1: In the paper “Regressive evolution of an effector following a host jump in the Irish Potato Famine Pathogen Lineage”, the authors the explore the evolution of the PexRD54 protein encoding gene within clade 1b and 1c of Phytophthora. The experiments are interesting and suitable, and the conclusions seem largely well supported, and provide an interesting new story on the evolution of these important pathogens. Overall, I enjoyed reading this thought-provoking paper.

Reviewer #2: The MS (PPATHOGENS-D-22-00023) is a continuation of research from this group that studies host jump and RXLR effector PexRD54 of Phytophthora infestans. Previously, they presented that a single polymorphism in the particular effectors may undergo adaption to host environments in the cases of Phytophthora infestans or Magnaporthe oryzae and other similar clade pathogens. They have also shown that PiPexRD54 may interact with plant ATG8 protein family via its C-terminal AIM sequence to interfere with the normal operation of host autophagy pathway. In this MS, the group presents PiPexRD54 from P. infestans and PmPexRD54 from P. mirabilis as another example for host specificity, and shows that a single amino acid polymorphism at AIM exhibit different binding activities with ATG8s from their corresponding host plants. Interestingly, PmPexRD54 AIM peptide bind very weakly to the M. jalapa ATG8s. Based on these results, the authors propose a model of regressive evolution of an effector in host jump. The MS is very well prepared with clear results and well-designed experiments. I like this kind of short and significant story very much.

Although the authors' concept is quite interesting and creative, some experiments are essential. I take some as examples.

1.What is the real function of PmPexRD54 in P. mirabilis infection? Transient expression of PiPexRD54 and PmPexRD54 in Nicotiana benthamiana is desirable.

2.Does PmPexRD54 also interfere with plant autophagy pathway, which is comparable with PiPexRD54 in previous reports or not?

3.As their previous papers, any amino acid substitution except Pro at position 379 does not alter interaction PexRD54 with ATG8s. It is possible to predict and compare the structure modeling of interaction between PmPexRD54-ATG8s and PiPexRD54-ATG8s?

4.How about other RXLR effectors with AIM in P. mirabilis? Does it have?

Reviewer #3: Host jump is a major model that pathogens expand or switch their host range. However, the process of pathogen–host specialization is remaining largely unknown. In this manuscript, the authors identified the AIM motif in the effector PexRD54 has degenerated following a host jump in a Phytophthora lineage”. This is an interesting story. The authors showed a good example that how pathogen evolves by losing molecular function to adapt new host environment. To strengthen the manuscript, I have some comments here.

**Part II – Major Issues: Key Experiments Required for Acceptance**

Reviewer #1: I don’t think any of the issues I have are major (apart from perhaps the control in the co-IP suggestion that can perhaps be reasonably explained), and should mostly be reasonably quick to address either in the wording/text. I will therefore list them all below in the ‘minor issues’.

Reviewer #2: Same as above.

Reviewer #3: (No Response)

**Part III – Minor Issues: Editorial and Data Presentation Modifications**

Reviewer #1: Figure S1 is currently not a very informative phylogenetic tree. The legend does not specify how the tree was constructed, there is no scale bar, or indication of what the branch lengths show, and no details of confidence or support such a bootstraps. They mention its taken from another paper, but all of this information should be briefly mentioned in the legend, and described in further detail in the “Phylogenetic analyses” methods section of this paper. I think these are important details for interpreting every tree, but are particularly important given the very small branch lengths between the species being discussed.

“Of these species, P.infestans ... evolved from a series of recent host jumps”. I have two questions about this: 1) Is there any dating done to ascertain the timing, or how do you know it is recent (and what is meant by recent)? It would be good to provide further details about what is known regarding this. 2) Given that some species of clade 1b and 1c have a broad host-range, as mentioned in Fig S1 and elsewhere, is it not possible that the ancestor also had a broad-host range, and that some of these current pathogens have specialised, rather than jumped host? Can the authors provide further information to clarify this point? And potentially clarify if necessary throughout (intro, discussion etc).

“possibly due to sequencing errors” – such as? Why have the PexRF54 sequence for P. paristica been taken from transcriptome data? Is there no genome assembly for it? In addition to causing a truncation, might there be other issues such as splicing, mis-assembly or reduced quality? Might this prove an issue for the downstream analysis and interpretation of this sequence?

“This data falsified our initial hypothesis, and we conclude.”. A very minor thought about this, is a concern about writing your a priori hypothesis in this paragraph as a central topic (starting and finishing the paragraph). It might be better to simply state what the results showed or suggested, rather than what you thought was going to happen before the experiment.

Regarding the co-IP, was there a reason for not including P. ipomoea or P. infestans PexRD54 (that do have AIM predicted domains) and their ATG8 targets? (to make sure the experiment is working as expected), as you have done so with the ITC experiment.

In the discussion, the authors interestingly discuss the other biochemical roles that have been found for PexRD, and correctly suggest that by nature of it being evolutionary conserved, it is functional, and indeed evidence that it is expressed and localised in a similar manner across hosts supports that. But a main argument (or at least phrase) is it’s an example of regressive evolution (in this case reduced ability to interact with ATG8). Given PexRD has a range of biochemical roles, could the loss of ATG8 binding co-inside with increased ability to perform one of those other roles the authors discuss, or perhaps even another unknown role? The word ‘regressive’ used throughout seems a little loaded given its defined as ‘returning to a former or less developed state’ – while presumably, this change is advantageous to P. mirablis. I imagine a gene swap experiment that shows P. mirablis is better able to infect its host using another species PexRD would support the notion its PexRD mutation(s) are actually regressive. I think some of these points would be good to clarify – and potentially rephrased.

Reviewer #2: Same as above.

Reviewer #3: 1. The authors conclude that the functionality of the PexRD54 AIM was lost in the P. mirabilis lineage, but does mutated PmPexRD54 contribute to virulence? For example, whether overexpress PmPexRD54 on M. jalapa have contribution to infection of P. mirabilis? Alternatively, does PmPexRD54 impair MjATG8s mediated autophagy function in Mirabilis plant? Do authors test anything on this?

2. Proposing the regressive evolution by using P. mirabilis PexRD54 AIM sequence example is fine. Whether mutated PexRD54 gain other function remain unclear and how many similar examples are not sure. There are several things are not sure. I suggest authors to shorten the speculation part in the manuscript.

3. In S3 Fig, the transcription pattern of PmPexRD54 are different in three RNA-seq repeats, two of them showed that PmPexRD54 was not expressed in the early stage of infection(< 5dpi). One may ask the consistency of the experiment. It’s better to show other Phytophthora genes as control and make a good explanation.

4. Please update the references 10, “A single amino acid polymorphism in a conserved effector of the multihost blast fungus pathogen expands host-target binding spectrum.” has been published on PLOS Pathogens.

PLOS authors have the option to publish the peer review history of their article (what does this mean?). If published, this will include your full peer review and any attached files.

Reviewer #1: No

Reviewer #2: No

Reviewer #3: No

Figure Files:

Data Requirements:

Reproducibility:

References:

---

## [Decision Letter · Decision Letter 1]

5 Oct 2022

Dear Prof. Kamoun,

We are pleased to inform you that your manuscript 'Regressive evolution of an effector following a host jump in the Irish Potato Famine Pathogen Lineage' has been provisionally accepted for publication in PLOS Pathogens.

Best regards,

Yuanchao Wang

Associate Editor

PLOS Pathogens

Hui-Shan Guo

Section Editor

PLOS Pathogens

Kasturi Haldar

Editor-in-Chief

PLOS Pathogens

orcid.org/0000-0001-5065-158X

Michael Malim

Editor-in-Chief

PLOS Pathogens

orcid.org/0000-0002-7699-2064

Reviewer Comments (if any, and for reference):

Reviewer's Responses to Questions

**Part I - Summary**

Reviewer #2: I recommend to accept this ms because of its significant and interesting findings. Honestly to say, it lacks several experiments to support their conclusion, which have been raised by the reviewers but were dismissed by the authors.

Reviewer #3: In this revised manuscript, authors addressed my previous concerns and I think this version meets the publication standard of PLoS pathogens.

**Part II – Major Issues: Key Experiments Required for Acceptance**

Reviewer #2: It is desirable to know whether PmPexRD54 contributes to virulence and interferes with plant autophagy process using transient expression assay in N. benth. I concern this because some other possibilities may exist, including functional diversity in evolution, by which PmPexRD54 may also counter autophagy-mediated defense by interacting with other components instead of ATG8.

Reviewer #3: I am satisfying about the answers from authors. Although authors can not supply more data, but I recognize their explanation that the manipulation on pathogens in this system is difficult and the transcriptional pattern are sometimes not always consistent in different assays. Authors are honest with the raw data and reviewer should not be critical on this.

**Part III – Minor Issues: Editorial and Data Presentation Modifications**

Reviewer #2: The ms has been well presented and it is easily to be followed.

Reviewer #3: Satisfying with the answer.

PLOS authors have the option to publish the peer review history of their article (what does this mean?). If published, this will include your full peer review and any attached files.

Reviewer #2: No

Reviewer #3: No

---

## [Editor Report · Acceptance letter]

23 Oct 2022

Dear Kamoun,

We are delighted to inform you that your manuscript, "Regressive evolution of an effector following a host jump in the Irish Potato Famine Pathogen Lineage," has been formally accepted for publication in PLOS Pathogens.

Best regards,

Kasturi Haldar

Editor-in-Chief

PLOS Pathogens

orcid.org/0000-0001-5065-158X

Michael Malim

Editor-in-Chief

PLOS Pathogens

orcid.org/0000-0002-7699-2064